# MixedSCNet: LiDAR-Based Place Recognition Using Multi-Channel Scan Context Neural Network

Yan Si [ID], Wenyi Han [ID], Die Yu, Baizhong Bao, Jian Duan [ID], Xiaobin Zhan [ID] and Tielin Shi *[ID]

School of Mechanical Science and Engineering, Huazhong University of Science and Technology, Wuhan 430074, China; yansi@hust.edu.cn (Y.S.); wyhan@hust.edu.cn (W.H.); yudie@hust.edu.cn (D.Y.); baobaizhong@hust.edu.cn (B.B.); zhanxb@hust.edu.cn (X.Z.)
* Correspondence: tlshi@hust.edu.cn

**Abstract:** In the realm of LiDAR-based place recognition tasks, three predominant methodologies have emerged: manually crafted feature descriptor-based methods, deep learning-based methods, and hybrid methods that combine the former two. Manually crafted feature descriptors often falter in reverse visits and confined indoor environments, while deep learning-based methods exhibit limitations in terms of generalization to distinct data domains. Hybrid methods tend to fix these problems, albeit at the cost of an expensive computational burden. In response to this, this paper introduces MixedSCNet, a novel hybrid approach designed to harness the strengths of manually crafted feature descriptors and deep learning models while keeping a relatively low computing overhead. MixedSCNet starts with constructing a BEV descriptor called MixedSC, which takes height, intensity, and smoothness into consideration simultaneously, thus offering a more comprehensive representation of the point cloud. Subsequently, MixedSC is fed into a compact Convolutional Neural Network (CNN), which further extracts high-level features, ultimately yielding a discriminative global point cloud descriptor. This descriptor is then employed for place retrieval, effectively bridging the gap between manually crafted feature descriptors and deep learning models. To substantiate the efficacy of this amalgamation, we undertake an extensive array of experiments on the KITTI and NCLT datasets. Results show that MixedSCNet stands out as the sole method showcasing state-of-the-art performance across both datasets, outperforming the other five methods while maintaining a relatively short runtime.

**Keywords:** LiDAR; place recognition; point cloud descriptor; BEV; CNN



## 1. Introduction

Global localization is an essential problem in autonomous navigation and is used to determine a robot's current location without any prior pose information [1,2]. Before performing global localization, it is common practice to first build a map of the environment and then match the collected sensor data with the map to determine the robot's current pose with respect to the map. Generally, global localization is split into two consecutive phases: place recognition and pose estimation [3]. The purpose of place recognition is to provide an initial location estimate, while pose estimation aims to calculate the robot's precise pose. Specific methods for global localization fall into two major categories. The first category couples place recognition and pose estimation, directly estimating the accurate pose of the robot [4–8]. The second one involves a two-stage approach, where place recognition provides a coarse location estimate and pose estimation refines this estimate to determine the precise pose [9–11]. For outdoor scenarios, Global Navigation Satellite Systems (GNSSs) are often used to provide an initial estimate of global location. However, in areas where GNSS signals are weak or unreliable, such as indoors, tunnels, or remote mountainous regions, a place recognition algorithm is a more generic and practical solution for a variety of environments.

Place recognition is essentially a retrieval problem, where the retrieval database is established during the processes of Simultaneous Localization and Mapping (SLAM) or Structure from Motion (SfM). The retrieval items within this database consist of global descriptors of keyframes. These global descriptors can be obtained by aggregating local descriptors [5,12] or directly extracting a single global descriptor from the original point cloud or image [3,13]. Each global descriptor is linked to the pose of its corresponding keyframe. Hence, when the global descriptor of the current frame is queried against the retrieval database, the keyframe exhibiting the highest similarity is identified as the nearest place. The pose associated with this keyframe can then be utilized as the initial pose estimate for the current frame.

In contrast to visual place recognition methods, LiDAR-based approaches offer two prominent advantages. Firstly, they are immune to significant variations in lighting conditions [3]. Secondly, LiDAR sensors boast a wide 360° field of view and the ability to acquire precise depth information. This enables LiDAR sensors to accurately capture the overall topological structure of the environment, rendering them less susceptible to local environmental fluctuations. As a result, LiDAR-based place recognition methods have gained growing interest in recent years.

Two representative methodologies in LiDAR-based place recognition are Bird's Eye View (BEV)-based models [6,13–17] and deep neural network-based models [3,11,18–21]. BEV descriptor-based methods initially transform point cloud data into two-dimensional representations. This transformation is accomplished by either projecting the point cloud onto a 2D plane and subsequently extracting 2D features [16] or extracting features from the point cloud first and then projecting them onto a 2D plane [15]. Successively, image-matching algorithms are employed for place retrieval. However, these methods may exhibit limitations in reverse visit situations and narrow environments [13]. Conversely, deep neural network-based methods directly take point cloud data as input and produce a point cloud descriptor as output [3]. Nevertheless, this approach comes with certain disadvantages. Firstly, it requires the downsampling of the point cloud data, as it cannot directly process the complete point cloud. Furthermore, it may demonstrate relatively poor generalization capabilities, particularly when the data distribution in the test set deviates from that in the training set, resulting in a noticeable decrease in performance.

In this paper, we propose MixedSC, which leverages the concept of Scan Context to divide point cloud data into multiple bins along radial and azimuthal directions in the Bird's Eye View. Within each bin, it records the maximum height, maximum intensity, and maximum smoothness, creating a three-channel tensor representation of the current point cloud, with each channel being represented as a fixed-size matrix. Subsequently, the MixedSC descriptor is fed into a compact deep neural network to generate a global descriptor enriched with high-level features. This descriptor is then employed in KD tree retrieval to identify the nearest keyframe. The primary contributions of our proposed method are as follows:

- A strongly discriminative point cloud descriptor, MixedSC. The configurations of the height channel and intensity channel exhibit high similarities in their shapes, while the smoothness channel records the local structure, thus making a difference, which enables robust matching even in reverse visit situations.
- An efficient feature extractor, MixedSCNet. By combining MixedSC and neural networks, it yields strong generalization in a distinct data domain with a relatively low computational overhead.
- Thorough experimental evaluations on the KITTI [22] and NCLT [23] datasets. Comparative assessments against five representative LiDAR-based place recognition methods, PointNetVLAD [3], Scan Context [13], Intensity Scan Context [17], MinkLoc3Dv2 [24], and BoW3D [5], demonstrate that our proposed approach achieves state-of-the-art performance and comparatively excellent generalization alibity.

## 2. Related Work

In recent years, research in the visual community has primarily focused on addressing the problem of place recognition using deep neural network-based methods, with a particular emphasis on multi-scales, attention mechanisms, and feature integration. Among these, Patch-NetVLAD [25] is noteworthy for its pioneering use of local–global patch-level descriptors across multiple scales. TransVPR [26], on the other hand, builds on the vision transformer framework, integrating attention across multiple levels to generate a global image representation. In contrast, MixVPR [27] introduces a holistic feature aggregation technique that treats feature maps from different pre-trained backbones as a global feature set. It then employs a cascade of feature mixing operations to capture global relationships between elements among each feature map, thus avoiding the local or pyramidal aggregation strategies employed in methods like NetVLAD or TransVPR.

In contrast to visual place recognition, LiDAR-based place recognition methods have garnered increasing attention in recent years for their insensitivity to lighting variations and capability to provide precise depth information. LiDAR-based place recognition methods can be categorized into three main groups: methods based on manually crafted feature descriptors, methods based on deep learning, and methods that combine both, all of which have demonstrated competitive performance.

### 2.1. Manually Crafted Feature Descriptor-Based Methods

The most critical aspect of methods based on manually crafted feature descriptors is the design of these descriptors, with Bird's Eye View (BEV) descriptors being the most commonly used ones in recent research. Scan Context [13] is one of the earliest methods to represent 3D LiDAR point clouds using BEV descriptors. It divides the space into $N_s$ sectors in the circumferential direction and $N_r$ rings in the radial direction, where the intersection area between a sector and a ring is called a bin. Scan Context records the height of the highest point within each bin, transforming a single LiDAR point cloud frame into a BEV descriptor matrix. Intensity Scan Context [17] enhances this by incorporating the reflection intensity information, making the descriptor more robust. Ref. [9] establishes a local reference frame by rotating the original LiDAR coordinate axis around the z-axis. In this frame, the x-axis corresponds to the direction of maximum point cloud variance, increasing the robustness of Scan Context to changes in viewpoint. BVMatch [16] uses the point number within each grid to represent that grid, constructing a BV image. Fast corner detection is then employed for feature extraction, with the detected keypoints being used for subsequent registration. Cross-Section Shape Context (CSSC) [10] is a descriptor similar to Scan Context, based on which Selective Generalized Iterative Closest Point (SGICP) is proposed for accurate pose estimation. Scan Context++ [14] introduces two main improvements over Scan Context: increased robustness to lateral motion and the ability to provide a one-degree-of-freedom initial pose estimate (yaw or lateral). Ref. [8] utilizes the Radon sinogram (RING) to characterize LiDAR scans, achieving robustness against arbitrary orientations and large displacements. It also presents a global convergent solver based on RING properties, addressing place recognition, rotation angle estimation, and displacement estimation in a single algorithm. BoW3D [5] initially extracts LinK3D descriptors from each frame's point cloud and constructs a hash table-like dictionary. It then employs a Bag-of-Words model to complete the place recognition task. Contour Context [6] proposes that scenes in BEV can be described by the probability distribution of salient structures. This method slices BEV images at different heights, creating levels, and obtains contours from each level, where each contour contains 2D structural information from a specific layer and is compressed into an ellipse and other parameters. Then, the similarity between two BEVs is computed based on the correlations among contours.

### 2.2. Deep Learning-Based Methods

Place recognition methods based on deep learning take point clouds as input to a neural network, which subsequently outputs a global descriptor. PointNetVLAD [3] was

the first place recognition method to directly input point clouds into a neural network. It initially utilizes PointNet to extract features for each point, mapping the point cloud from three dimensions to higher dimensions. Subsequently, it employs NetVLAD to identify K local feature clusters and output a VLAD global descriptor. Finally, a fully connected layer is used to reduce the dimensionality of the VLAD descriptor, yielding the ultimate global descriptor. LPD-Net [28] introduces an adaptive local feature extraction module, followed by concatenating the extracted local features of the point cloud with the transformed coordinates of the point cloud itself to create a mixed local feature. Furthermore, it employs a graph-based neighborhood aggregation module in both the feature space and cartesian space to reveal the spatial distribution of the mixed local features. PCAN [29] is a neural network with an attention mechanism that can predict the importance of each local point feature based on the point context, which helps the network focus more on task-relevant features when aggregating local features. Pyramid Point Cloud Transformer [18] introduces a pyramid point transformer module to adaptively learn the spatial relationships of different k-NN point clouds. It employs grouped self-attention to extract informative local features. Additionally, it uses a pyramid VLAD module to aggregate multi-scale feature maps. By performing VLAD pooling on multi-scale feature maps, it utilizes a context gating mechanism to adaptively assign weights to multi-scale global context information, resulting in a final global descriptor. MinkLoc3D [19] is based on a sparse voxelized point cloud representation and sparse 3D convolution to compute a discriminative global descriptor. SOE-Net [21] maximizes the relationships among points and incorporates long-range context into point-wise local descriptors. It extracts local information for each point from eight-directional local information using the PointOE module and extracts long-range feature dependencies among local features using a self-attention unit. LCDNet [4] consists of a shared encoder, a place recognition head, and a relative pose head. The place recognition head is used to extract global descriptors, while the relative pose head computes the relative pose between two frames of point clouds. SVT-Net [30] is a super lightweight model based on 3D Sparse Convolution (SP-Conv). It introduces an Atom-based Sparse Voxel Transformer (ASVT) and a Cluster-based Sparse Voxel Transformer (CSVT), which are used for learning short-range local features and long-range contextual features, respectively.

### 2.3. Hybrid Methods

Combining manually crafted feature descriptors with deep learning methods often yields superior performance, albeit at the cost of increased complexity. Ref. [31] processes Scan Context to create a Scan Context Image (SCI), which is then used to train a classification network, treating each place as a distinct class. Ref. [11] initially represents the distance range and differences in neighboring point distances in a histogram, which exhibits rotational invariance. This histogram is then input to a Siamese neural network for training, resulting in LocNet, a network capable of comparing the similarity of different point clouds. Semantic Scan Context [15] first performs semantic segmentation using RangeNet++. It projects representative object point clouds onto the x–y plane and aligns the 2D point clouds using global semantic ICP; then, it extracts Scan Context with semantic information from the aligned point clouds. In [7], SPI (Submap Projection Image) is proposed to project point clouds onto a horizontal plane, preserving the height of the highest point in the corresponding vertical bin for each pixel. Subsequently, SPI-NetVLAD, a neural network combining SuperPoint and SuperBlue, takes SPI as input to output a global descriptor. In [32], point clouds are transformed into Bird's Eye View (BEV) images and further converted into polar coordinates, resulting in a polar BEV image. The polar BEV image is input to a U-Net to obtain feature maps in polar coordinates. Finally, Fourier transformation is applied to transform the feature map into the frequency domain, generating a feature descriptor for the point cloud. Locus [20] initially extracts segments from point clouds. It then utilizes structural appearance and topological and temporal information as two features for these segments. Second-Order Pooling (O2P) and Power-Euclidean transform (PE) are applied to aggregate these features, resulting in a global descriptor.

OverlapTransformer [33] combines OverlapNet, Transformer, and NetVLAD to map the range image of a point cloud into a yaw-angle-invariant global descriptor. RINet [34] encodes the semantic and geometric information of a point cloud into a rotation-equivariant descriptor. Then, a Siamese neural network with modified convolution and pooling layers is used to ensure strict rotational invariance. CVTNet [35] integrates Range Image Views (RIVs) and Bird's Eye Views (BEVs). In CVNet, intra-transformers extract correlations within the same view, while inter-transformers extract correlations among different views, generating a yaw-angle-invariant global descriptor.

For LiDAR-based place recognition tasks, manually crafted feature descriptors often exhibit limited adaptability to narrow environments, while deep learning-based place recognition methods tend to suffer from poor generalization in out-of-training-set data domains, particularly for reverse loop cases. Furthermore, hybrid methods that combine manually crafted descriptors and deep learning often involve multiple steps, and there is a lot of room for improvements in processing speed. In this paper, we propose a LiDAR-based place recognition method utilizing Bird's Eye View (BEV) descriptors and lightweight neural networks. This method achieves rapid place recognition with reduced computational load, exhibits good adaptability to narrow environments, and demonstrates strong generalization capabilities in cross-domain and reverse loop scenarios.

## 3. Methodology

### 3.1. System Framework

In Figure 1, the system framework for our proposed method is illustrated. The overall process consists of two stages: database construction and place recognition. During the first stage, point clouds from the preceding keyframes are transformed into global descriptors through MixedSCNet, which are then organized in a KD tree to build the database. Subsequently, during the place recognition phase, the current frame's point cloud undergoes the same transformation to generate its corresponding global descriptor, which is later matched against those held by the database, yielding multiple candidate frames with the highest similarity scores.

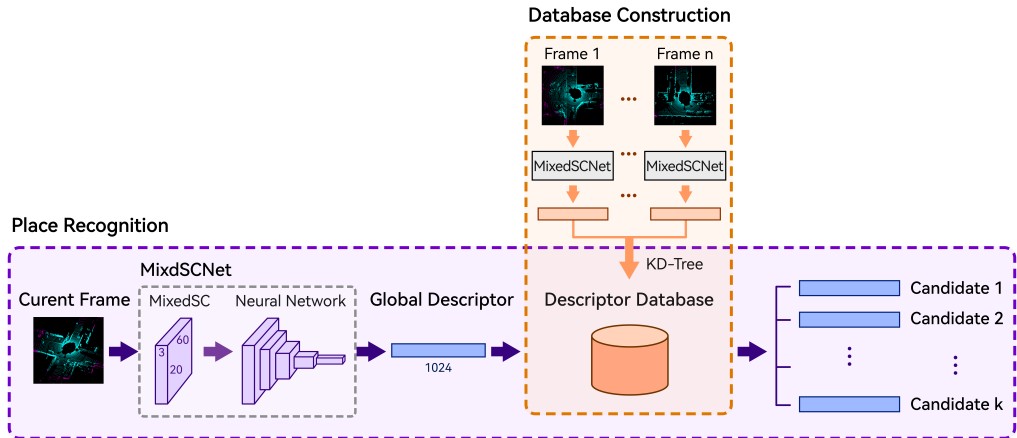

**Figure 1.** System framework of the proposed method.

### 3.2. Smoothness Calculation

The smoothness of a given point is defined as the difference in horizontal polar distance between this point and the average value of its neighboring points. It reflects the fluctuation in the depth direction near that point. Higher smoothness suggests more dramatic depth variation, which means the point tends to lie on an object featuring protrusions or depressions in physical space. Therefore, the distribution pattern of smoothness values across a point cloud effectively captures local geometric features, particularly those related to objects exhibiting sharp corners or edges.

Suppose that $\mathcal{P}_{src} = \{p_1, \ldots, p_n \mid p_n \in \mathbb{R}^4\}$ is a LiDAR point cloud, where $(x_k, y_k, z_k)$ is the Cartesian coordinates of the point $p_k$. First, express $p_k$ in a form analogous to spherical coordinates:

$$p_k = [r_k, \alpha_k, \omega_k, \eta_k], \tag{1}$$

$$r_k = \sqrt{x_k^2 + y_k^2}, \tag{2}$$

$$\alpha_k = arctan \frac{y_k}{x_k}, \tag{3}$$

$$\omega_k = arctan \frac{z_k}{r_k}, \tag{4}$$

where $r_k$, also known as the range, denotes the horizontal polar distance of point $p_k$; $\alpha_k$ is the azimuth angle, and $\omega_k$ is the elevation angle, with both being expressed in degrees; $\eta_k$ represents the reflection intensity. When considering a LiDAR with $l$ laser channels and $\theta$ as its angular resolution, there are $l \times \left(\frac{360}{\theta}\right)$ laser beams emitted by the LiDAR during a single revolution. For any point beyond the LiDAR's maximum or minimum detection range, its corresponding range, $r_k$, is set to zero. Consequently, these disparate range values coalesce into a matrix denoted by $R \in \mathbb{R}^{l \times \left(\frac{360}{\theta}\right)}$, termed as the range image. This mapping enables us to represent complex scenes captured using LiDAR using a simple yet powerful image format. Given a non-zero value $r_k$, it can be designated to a specific position $(i, j)$ within matrix $R$, i.e., $r_k = R_{ij}$. The index $(i, j)$ can be calculated as

$$i = \phi(\omega_k), \tag{5}$$

$$j = \left[\frac{\alpha_k}{\theta}\right], \tag{6}$$

where variable $i$ denotes that point $p_k$ corresponds to the result obtained from the $i$-th laser channel of the LiDAR sensor, the mapping relationship between $\omega_k$ and the laser channel index is represented by $\phi(\omega_k)$, and the notation $[\cdot]$ indicates rounding to the nearest integer. The function $\phi$ varies depending on the specific LiDAR model. For instance, for the Velodyne VLP16 LiDAR model, $\phi(\omega_k) = \left[\frac{\omega_k}{2} + 8\right]$.

Based on range image $R$, it is possible to compute the smoothness ($s_k$) of point $p_k$. Suppose that the range value of point $p_k$ is denoted by $r_k = R_{ij}$. The left and right neighborhoods of $r_k$ are defined as follows:

$$\mathbb{N}_k^- = \{R_{i,j-q} \mid q \in [|1, 5|]\}, \tag{7}$$

$$\mathbb{N}_k^+ = \{R_{i,j+q} \mid q \in [|1, 5|]\}, \tag{8}$$

where $[|1, N|]$ denotes the set $\{1, 2, \ldots, N\}$. Let $\mathbb{N}_k = \mathbb{N}_k^- \cup \mathbb{N}_k^+$; then, the expression for calculating $s_k$ can be formulated as follows:

$$s_k = \begin{cases} \left| \dfrac{\sum\limits_{r \in \mathbb{N}_k} r}{\|\mathbb{N}_k\|_0} - r_k \right|, & \text{if } \|\mathbb{N}_k^-\|_0 \geq 2, \|\mathbb{N}_k^+\|_0 \geq 2 \\ 0, & \text{otherwise} \end{cases} \tag{9}$$

where the notation $\|\cdot\|_0$ denotes the number of non-zero elements.

### 3.3. Mixed Scan Context

Inspired by Scan Context [13] and Intensity Scan Context [17], this paper introduces a highly discriminative Bird's Eye View (BEV) point cloud descriptor termed Mixed Scan Context (MixedSC). Scan Context captures the highest height values in the local point cloud, aggregating them into a matrix that serves as the global descriptor for the entire point cloud. This effectively summarizes the point cloud's contour structure. However,

this approach exhibits limitations in narrow environments, where the height variations in LiDAR point clouds are not pronounced. Intensity Scan Context, on the other hand, selects the highest intensity values in the local point cloud. Yet, due to the distance-dependent nature of intensity, variations may occur for the same object under different observation positions and angles. Additionally, environmental factors, such as weather conditions and surface chemical reactions over time, can influence object surface characteristics, thereby impacting intensity values. Hence, relying solely on intensity-based descriptors lacks sufficient robustness. To address these limitations, this paper proposes MixedSC, which combines height, intensity, and smoothness information. By incorporating point cloud contour, object surface characteristics, and local geometric features, MixedSC shows to be a strong global descriptor to effectively accommodate narrow environments, fluctuating weather conditions, and variations in object surface properties.

Initially, it retains the height, range, azimuth angle, intensity, and smoothness information of a point cloud and filters out points within specified range and height intervals:

$$\mathcal{P} = \{p_k = [z_k, r_k, \alpha_k, \eta_k, s_k] \in \mathcal{P}_{src} \mid r_{min} \leq r_k \leq r_{max}, z_{min} \leq z_k \leq z_{max}\}. \quad (10)$$

Subsequently, as illustrated in Figure 2, akin to Scan Context, MixedSC divides the space into $N_s = 60$ sectors azimuthally and $N_r = 20$ rings radially. The intersecting region between each sector and ring is referred to as a bin. Let the set of point clouds within the intersecting region of the $m$-th ring and $n$-th sector be denoted by $\mathcal{P}_{mn}$, assuming that $p_k \in \mathcal{P}_{mn}$. Then,

$$m = \left\lfloor \frac{r_k - r_{min}}{r_{max} - r_{min}} \cdot N_r \right\rfloor, \quad (11)$$

$$n = \left\lfloor \left( \frac{\alpha_k}{360} + \frac{1}{2} \right) \cdot N_s \right\rfloor, \quad (12)$$

where $\lfloor \cdot \rfloor$ denotes the floor function. By utilizing Formulas (11) and (12), each point is assigned to its corresponding bin. Afterward, employing the bin encoding function $\psi : \mathcal{P}_{mn} \to \mathbb{R}^3$, the highest height value, the maximum reflectance intensity value, and the highest smoothness value within each bin are recorded:

$$\psi(\mathcal{P}_{mn}) = \left[ \max_{p_k \in \mathcal{P}_{mn}} z_k, \max_{p_k \in \mathcal{P}_{mn}} \eta_k, \max_{p_k \in \mathcal{P}_{mn}} s_k \right]. \quad (13)$$

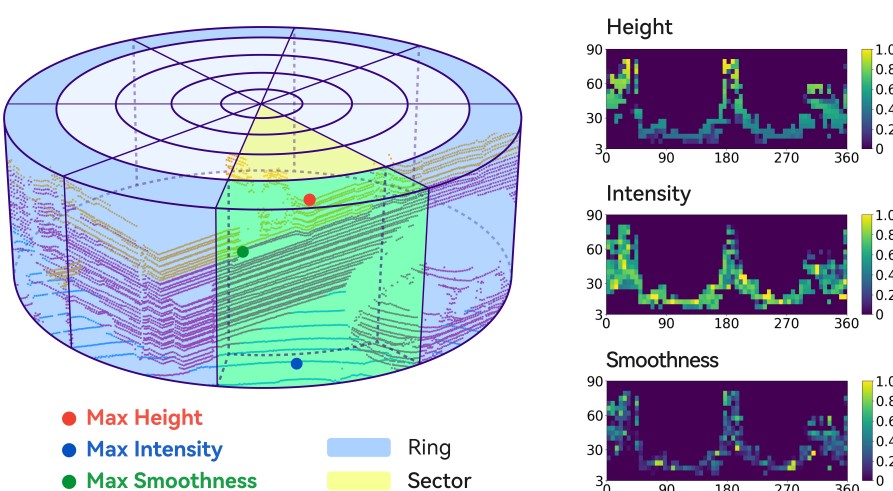

**Figure 2.** Visualization of MixedSC. The light-blue region corresponds to a ring; the pale-yellow area represents a sector; and the light-green area indicates the bin resulting from their intersection. The amalgamation of the three right-side matrices results in a MixedSC descriptor.

After the aforementioned process, a single frame of LiDAR point cloud is ultimately transformed into a tensor ($T$) called MixedSC, i.e.,

$$T = (t_{mn}) \in \mathbb{R}^{3 \times N_r \times N_s}, \; t_{mn} = \psi(\mathcal{P}_{mn}). \tag{14}$$

*3.4. MixedSCNet*

3.4.1. Network Architecture

The network architecture is illustrated in Figure 3. MixedSCNet is a compact Convolutional Neural Network similar to ResNet18 [36]. Given MixedSC as input, an initial convolution operation with a $5 \times 5$ kernel is performed, elevating the channel dimension to 64. Subsequently, a $3 \times 3$ max pooling layer is applied, downsampling the image to $10 \times 30$. In the subsequent main network, the convolutional kernel size remains $3 \times 3$, with $1 \times 1$ kernels in skip connections. The final step involves reducing the feature map dimensions to a 1024-dimensional global descriptor through an average pooling layer.

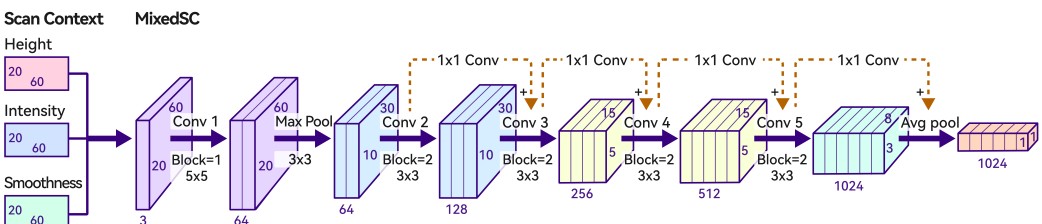

**Figure 3.** The network architecture of MixedSCNet. *Block* denotes the number of convolutional operations. $1 \times 1$, $3 \times 3$, and $5 \times 5$ represent kernel sizes.

Using a Convolutional Neural Network to process MixedSC is motivated by its effectiveness in extracting local features from images. MixedSC can be likened to a low-resolution 360° panoramic snapshot of the real-world scene, with its three channels exhibiting similar overall shapes, reflecting the approximate geometric structure of the scene. However, each channel's values differ from each other, as they represent three distinct features. Consequently, the data format of MixedSC resembles that of an RGB image. Moreover, the proposed Convolutional Neural Network has a relatively low computation overhead, totaling only $1.76 \times 10^9$ FLOPs, which ensures good performance and efficient computation speed at the same time.

3.4.2. Metric Learning

A training sample can be represented as a tuple $\mathcal{T} = (P_i, \mathcal{P}_{pos}, \mathcal{P}_{neg})$. $P_i$ represents the query cloud, and $o_i$ denotes its origin. $\mathcal{P}_{pos}$ and $\mathcal{P}_{neg}$ denote the set of similar frames of $P_i$, and the set of dissimilar frames of $P_i$ respectively, which are defined as

$$\mathcal{P}_{pos} = \left\{ P_j \,\middle|\, \left\| o_i - o_j \right\|_2 \leq D_{pos}, |i - j| > \Delta \right\}, \tag{15}$$

$$\mathcal{P}_{neg} = \left\{ P_j \,\middle|\, \left\| o_i - o_j \right\|_2 > D_{neg} \right\}, \tag{16}$$

where $D_{pos}$ represents the distance threshold between similar frames, $\Delta$ denotes the frame index difference threshold between similar frames, and $D_{neg}$ signifies the distance threshold between dissimilar frames. The loss function ($\mathcal{L}_{trip}$) employed in this study is based on the lazy triplet loss function from PointNetVLAD [3], which can be formulated as

$$\mathcal{L}_{trip}(\mathcal{T}) = \left[ \alpha + \max_i \left\| f(P_a) - f\left(\mathcal{P}_{pos}^i\right) \right\|_2 - \min_j \left\| f(P_a) - f\left(\mathcal{P}_{neg}^j\right) \right\|_2 \right]_+, \tag{17}$$

where $f(\cdot)$ represents the mapping function represented by the trained MixedSCNet, $\alpha$ is a constant, and $[\cdot]_+$ denotes the hinge loss. This loss function selects the positive cloud with

the maximum distance from the query cloud and the negative cloud with the minimum distance from the query cloud. Therefore, in each iteration, the neural network is optimized by selecting the pair of the most challenging positive cloud and negative cloud. This process aims to minimize the distance between the query cloud and the positive cloud while maximizing the distance from the negative cloud.

### 3.4.3. Data Augmentation

Inspired by [31], this paper similarly employs the column-shift data augmentation technique to enhance the robustness of MixedSCNet against variations in viewpoints. Specifically, the MixedSC tensor is initially partitioned into $N = 4$ sub-modules along the vertical axis, denoted by $T = [T_1, \ldots, T_N]$. Subsequently, these sub-modules undergo cyclic right shifts for $M$ times, where $M$ is randomly determined during the training process within the range $[|1, N|]$. This process yields a new MixedSC tensor, denoted by $T' = \left[ T'_1, \ldots, T'_N \right]$, and the correspondence between the sub-modules of the old and new MixedSC is defined as $T'_j = T_i$, where

$$
j = \begin{cases} (i + M) \ \% \ N, & \text{if } i + M \neq N \\ N, & \text{otherwise} \end{cases}
\tag{18}
$$

with % denoting the modulo operation. Consequently, during the training procedure, the network encounters MixedSC from different perspectives, enabling it to learn invariance to viewpoint changes.

## 4. Experiments

### 4.1. Dataset and Experimental Settings

The KITTI dataset, derived from the Karlsruhe Institute of Technology and Toyota Technological Institute, stands as a widely recognized benchmark in autonomous driving research. It comprises an extensive array of high-resolution sensor data collected with a mobile platform operating in urban environments. The dataset employs the Velodyne HDL-64E LiDAR module. Leveraging its real-world complexity and richness, the KITTI dataset serves as a valuable resource for evaluating the robustness and generalization capabilities of place recognition models within challenging urban settings. The KITTI odometry benchmark consists of 22 sequences, with sequences 0-10 providing ground-truth poses. Notably, sequences with loop closures are 00, 02, 05, 06, 07, and 08. Consequently, this study primarily conducts experiments on these six sequences. Specifically, sequences 00, 02, and 08 are used for training, while all sequences are designated for testing. The training set of the KITTI dataset comprises a total of 2465 samples, while the test set contains 1538 samples.

The North Campus Long-Term (NCLT) dataset consists of data from various sensors collected with a Segway robotic platform. Spanning 34.9 h of recorded logs and covering a robot trajectory of 147.4 km across 27 mapping sessions, this dataset provides a comprehensive representation of diverse environments, including both indoor and outdoor settings. The Velodyne HDL-32E LiDAR module is employed for data acquisition. This dataset offers an extensive and varied collection of sensor data captured over an extended duration. In this study, we randomly select five sequences, namely, 2012-02-02, 2012-02-04, 2012-02-05, 2012-05-26, and 2012-08-20, to conduct experiments. Importantly, these sequences are exclusively utilized to test the cross-domain generalization performance of models trained on the KITTI dataset. Starting from the first frame of each sequence, query clouds are sampled at every 1-meter interval. The NCLT dataset's test set comprises a total of 9216 samples.

The number of frames and test samples for each sequence is detailed in Table 1. In the training set, for a training sample denoted by $\mathcal{T}_{train} = (P_i, \mathcal{P}_{pos}, \mathcal{P}_{neg})$, the distance thresholds in Formulas (15) and (16) are set to $D_{pos} = 5$ m and $D_{neg} = 10$ m, while the frame

index difference threshold is $\Delta = 0$. The choice of $\Delta = 0$ is made to maximize the number of training samples, aligning it with the number of frames in each training sequence. In the test set, for a test sample denoted by $\mathcal{T}_{test} = (P_i, \mathcal{P}_{pos})$, the distance threshold for similar frames is $D_{pos} = 3$ m, and the frame index difference threshold is $\Delta = 300$. Setting $\Delta$ to 300 is intended to filter out several frames before and after the current frame. These frames generally exhibit minor environmental changes, since their timestamps are very close to the current frame. Incorporating such frames in the set of similar frames is deemed less meaningful. This adjustment aligns the place recognition task more closely with real-world scenarios, rendering it more challenging.

**Table 1.** The number of frames and test samples for each sequence.

| Sequence | KITTI | | | | | | NCLT | | | | |
|---|---|---|---|---|---|---|---|---|---|---|---|
| | 00 | 02 | 05 | 06 | 07 | 08 | 2012-02-02 | 2012-02-04 | 2012-02-05 | 2012-05-26 | 2012-08-20 |
| Frames | 4541 | 4661 | 2761 | 1101 | 1101 | 4071 | 5544 | 4938 | 5830 | 5598 | 5347 |
| Test samples | 1656 | 495 | 920 | 558 | 60 | 314 | 2415 | 1552 | 1674 | 2110 | 1465 |

Two GeForce GTX 1080 Ti (Nvidia, Wuhan, China) are deployed to train MixedSCNet with distributed training techniques. The training batch size is configured to 1, comprising a query point cloud, two randomly selected positive point clouds, and eighteen random negative point clouds. Experience suggests that setting the batch size to 1 makes the model less prone to getting trapped in local optima and makes it converge more effectively. Adam is the chosen optimizer, with a base learning rate set to $1 \times 10^{-3}$. When the top 1% recall does not improve for five consecutive epochs, the learning rate is reduced to 0.1 times the rate of the previous epoch, and the minimum learning rate is set to $1 \times 10^{-6}$. In Equation (10), $r_{min} = 3$ m and $r_{max} = 90$ m. Due to variations in the installation height and angle of the LiDAR sensor in the KITTI and NCLT datasets, the $z_{min}$ and $z_{max}$ values differ. For KITTI, $z_{min} = -0.9$ m, and $z_{max} = 3.2$ m; on the other hand, for NCLT, $z_{min} = -20.0$ m, and $z_{max} = 1.0$ m. In Equation (17), the margin $\alpha$ is set to 0.5.

*4.2. Comparison Methods*

This study conducts comparative analyses between MixedSCNet and five other methods: PointNetVLAD, Scan Context, Intensity Scan Context, MinkLoc3Dv2, and BoW3D:

(1) PointNetVLAD [3], proposed by Mikaela Angelina Uy et al. in 2018, is a deep neural network-based place recognition method that combines PointNet and NetVLAD. For the in-house datasets in the original paper, a $25 \times 25$ bounding box was split, and the points within the box were downsampled to the number of 4096 and then fed to the neural network. For the KITTI and NCLT datasets used in this study, we similarly define a bounding box centered at the origin of a single-frame point cloud and downsample to 4096 points, aligning the input format with the original paper. Subsequently, we utilize the refined pre-trained model in the official source code (https://github.com/mikacuy/pointnetvlad (accessed on 1 December 2023)) and retrieve candidate frames based on the global descriptor generated by PointNetVLAD.

(2) Scan Context [13], introduced by Giseop Kim et al. in 2018, is a place recognition method utilizing manually designed feature descriptors. We refer to its open-source implementation integrated with LIO-SAM [37], SC-LIO-SAM (https://github.com/gisbi-kim/SC-LIO-SAM (accessed on 15 November 2023)), whose default parameters are retained. Following the method in [13], we employ the ring key to retrieve several similar frames as candidate frames, accomplishing the place recognition task.

(3) Intensity Scan Context (ISC) [17] is an improved algorithm of Scan Context proposed by Han Wang et al. in 2020, incorporating both geometric structure and reflection intensity information. We reference the open-source implementation ISCLOAM (https://github.com/wh200720041/iscloam (accessed on 7 January 2024)), maintain-

ing default parameters and adhering to the same criteria for evaluating candidate frame similarity scores, where the total similarity score for candidate frames is computed as the sum of geometric and intensity scores.

(4)  MinkLoc3Dv2 [24], introduced by Jacek Komorowski et al. in 2022, represents a deep neural network-based place recognition method. At present, MinkLoc3Dv2 exhibits the best performance among all open-source implementations on the Oxford RobotCar dataset. The input data requirements for MinkLoc3Dv2 align with those of PointNetVLAD, necessitating the adoption of identical point cloud preprocessing methods during reproduction. We utilize the refined pre-trained models provided by the official source code (https://github.com/jac99/MinkLoc3Dv2) (accessed on 7 January 2024) and perform candidate frame retrieval based on the global descriptors generated by MinkLoc3Dv2.

(5)  BoW3D [5], presented by Yunge Cui et al. in 2023, is a loop closure detection method based on the handcrafted descriptor LinK3D [38]. The original paper only demonstrates loop closure detection results on KITTI, and this study extends the evaluation to place recognition tasks on two datasets. Given that the original paper focused on loop closure detection, the Bag of Words (BoW) is dynamically updated in real time. However, when applied to the place recognition task, we traverse the data twice. The first traversal is conducted to establish a comprehensive vocabulary database. In the second pass, where the database is no longer updated, candidate frames for each frame are identified within the database and subsequently arranged in descending order based on their associated scores. The entire experimental procedure maintains the default parameters as specified in the source code (https://github.com/YungeCui/BoW3D) (accessed on 20 November 2023).

### 4.3. Place Recognition Performance

As conducted in PointNetVLAD, we assess several methods on the KITTI dataset and the NCLT dataset based on the average recall rate at top 1 and top 1%. Notably, recall rate at top 1% holds greater practical relevance in real-world scenarios, as paths traversed by robots may encompass similar scenes. Strictly adhering to the top candidate in such situations may result in missed detections, thereby compromising the robustness of localization.

Table 2 presents the recall rate performance of MixedSCNet and five comparative methods. Due to environmental variations, an evident performance drop is observed for PointNetVLAD on both the KITTI and NCLT datasets compared with its recall rates on the three test sets in the original paper. Notably, the pronounced decline in recall rate on the NCLT dataset is attributed to its 32-line LiDAR sensor, in contrast to the 64-line LiDAR used in the original paper's test sets. Moreover, the superior performance of Scan Context with default parameters on KITTI can be attributed to the fine tuning of its parameters specifically for the KITTI dataset. However, a noticeable decline is observed on the NCLT dataset, primarily due to the inclusion of numerous narrow indoor environments, where Scan Context exhibits diminished adaptability. Intensity Scan Context, evaluated initially on the KITTI dataset, demonstrates the highest recall at top 1. The combination of geometric and intensity features, along with feature extraction from raw point clouds, contributes to its superior performance on KITTI. Nevertheless, a similar performance drop is observed on the NCLT dataset, akin to Scan Context. As the state-of-the-art method on the Oxford RobotCar dataset, MinkLoc3Dv2 maintains good generalization on NCLT, displaying the highest recall at top 1%. Its top 1 recall, however, slightly lags behind Intensity Scan Context and MixedSCNet. Notably, when transferred to the KITTI dataset, MinkLoc3Dv2 underperforms obviously compared with two manually designed BEV descriptor-based methods and MixedSCNet. BoW3D exhibits the poorest performance among the six methods. For a place recognition algorithm with high robustness, its performance in loop closure detection and place recognition tasks should ideally be consistent. However, in the original paper, BoW3D demonstrated excellent loop closure detection results on KITTI but considerably poor place recognition performance with the same set of hyperparameters

under this paper's task settings. Its performance further deteriorates when transferred to the NCLT dataset.

**Table 2.** Average recall (%) at top 1 and top 1% of different methods on KITTI and NCLT datasets. The bold number represents the highest recall among all the comparative methods in one dataset.

| Dataset | Method | Recall@1 | Recall@1% |
|---------|--------|----------|-----------|
| KITTI | PointNetVLAD | 65.95 | 81.41 |
| | Scan Context | 89.06 | 96.63 |
| | Intensity Scan Context | **92.21** | 96.98 |
| | MinkLoc3Dv2 | 71.82 | 93.50 |
| | BoW3D | 34.32 | 56.61 |
| | MixedSCNet (ours) | 90.28 | **99.30** |
| NCLT | PointNetVLAD | 24.40 | 41.06 |
| | Scan Context | 27.47 | 76.11 |
| | Intensity Scan Context | 68.50 | 86.35 |
| | MinkLoc3Dv2 | 67.95 | **99.15** |
| | BoW3D | 10.56 | 14.84 |
| | MixedSCNet (ours) | **69.61** | 98.98 |

In contrast, MixedSCNet is the only method that demonstrates state-of-the-art performance on both datasets. On the KITTI dataset, MixedSCNet and Intensity Scan Context stand as the top methods. Notably, MixedSCNet achieves the highest recall at top 1%, with a slightly lower recall at top 1 compared with Intensity Scan Context. On the NCLT dataset, MixedSCNet and MinkLoc3Dv2 emerge as the top performers, significantly surpassing alternative methods. Specifically, MixedSCNet achieves the highest recall at top 1, with recall at top 1% closely trailing MinkLoc3Dv2. Trained on three sequences of the KITTI dataset, MixedSCNet is expected to perform well on KITTI, but its remarkable performance on the NCLT dataset, with almost no decrease in recall rate at top 1% and the highest top 1 recall rate, indicates its exceptional cross-domain generalization capabilities. Additionally, MixedSCNet's performance on the NCLT dataset underscores its adaptability to narrow environments, as it retains intensity and smoothness information from point clouds, which proves to be effective in such confined spaces.

Figure 4 illustrates the importance of the smoothness channel. Frames 350 and 5333, extracted from sequence 2012-02-02 of the NCLT dataset, constitute a reverse visit. It can be observed that the similarity in the intensity channel of the two point clouds is weaker compared with the smoothness channel. Consequently, the Intensity Scan Context algorithm fails to include frame 5333 in the top 1% candidate frames of frame 350. In contrast, MixedSCNet successfully identifies it. Figure 5 showcases another pair of reverse visit frames from the same sequence. Intensity Scan Context also fails to correctly recognize it. However, the feature maps output by MixedSCNet reveal a minimal dissimilarity between the query frame and the reverse visit frame while maintaining a substantial dissimilarity with other negative frames.

Figure 6 illustrates the recall–candidate number curves for the six methods across all sequences. For the KITTI dataset, sequence 02 poses a notable challenge, exhibiting a considerable degradation in performance for all methods compared with other sequences. BoW3D, in particular, experiences a pronounced decline, while MixedSCNet displays the smallest performance drop among the six methods. Sequence 08 proves to be the most challenging for PointNetVLAD due to the substantial presence of reverse visits, which PointNetVLAD struggles to identify. Regarding the NCLT dataset, significant variations in scenes and LiDAR channel numbers lead to a marked decline in performance for the other three methods. In contrast, MixedSCNet demonstrates minimal deterioration in performance, indicating its robust cross-domain generalization capabilities.

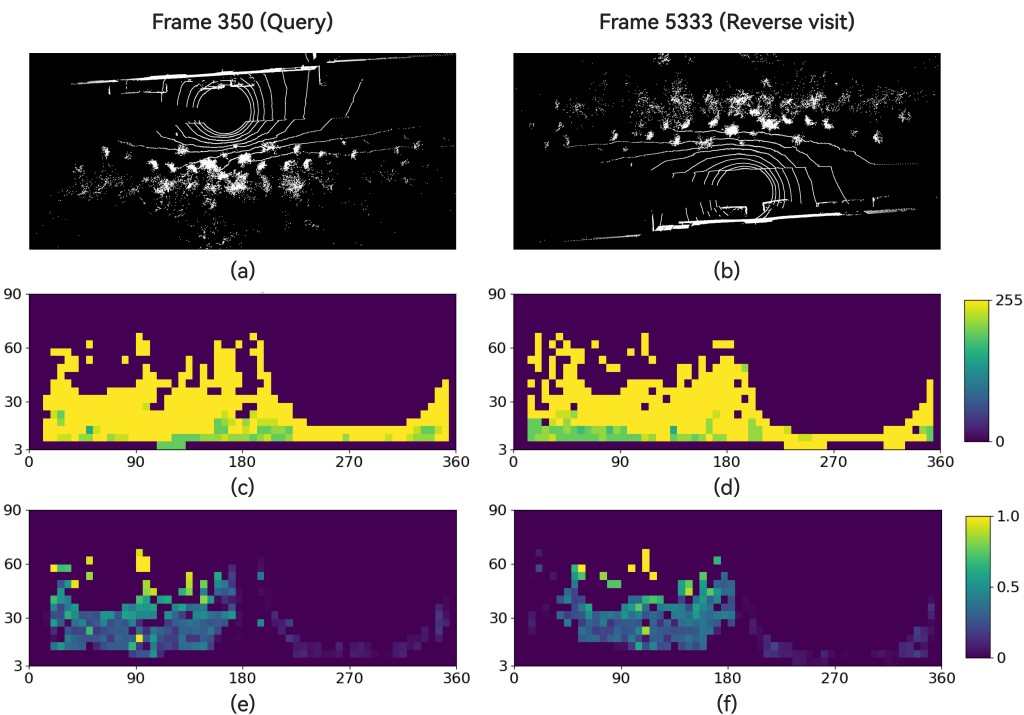

**Figure 4.** Intensity and smoothness channel visualization for a reverse visit case. (**a**,**b**) Reverse visit in NCLT 2012-02-02 sequence (**c**,**d**) and corresponding intensity channels produced by Intensity Scan Context, where the latter one is the 180°-shifted version of the source channel. (**e**,**f**) The smoothness channels of their corresponding MixedSC.

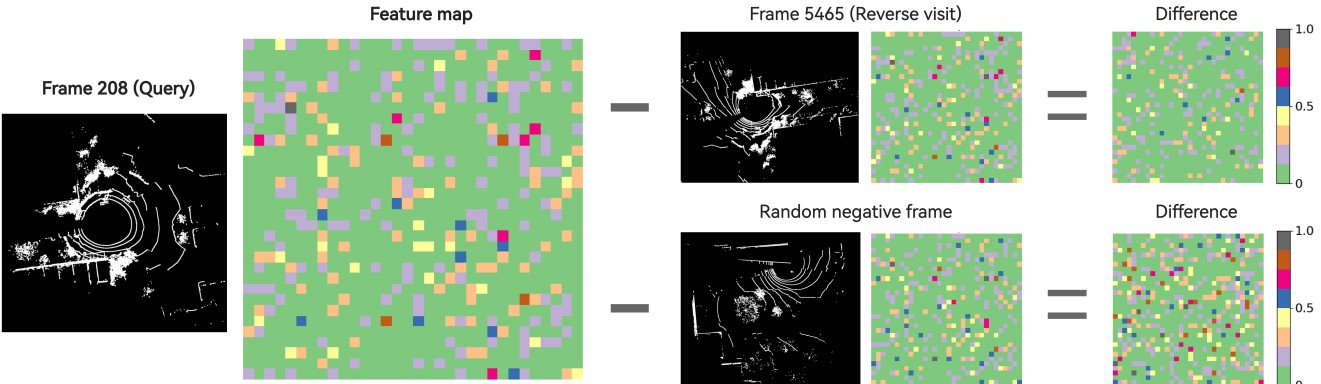

**Figure 5.** Feature map visualization for MixedSCNet. The feature vector generated by MixedSCNet is reshaped into a 32 × 32 matrix, which stands for the final feature map.

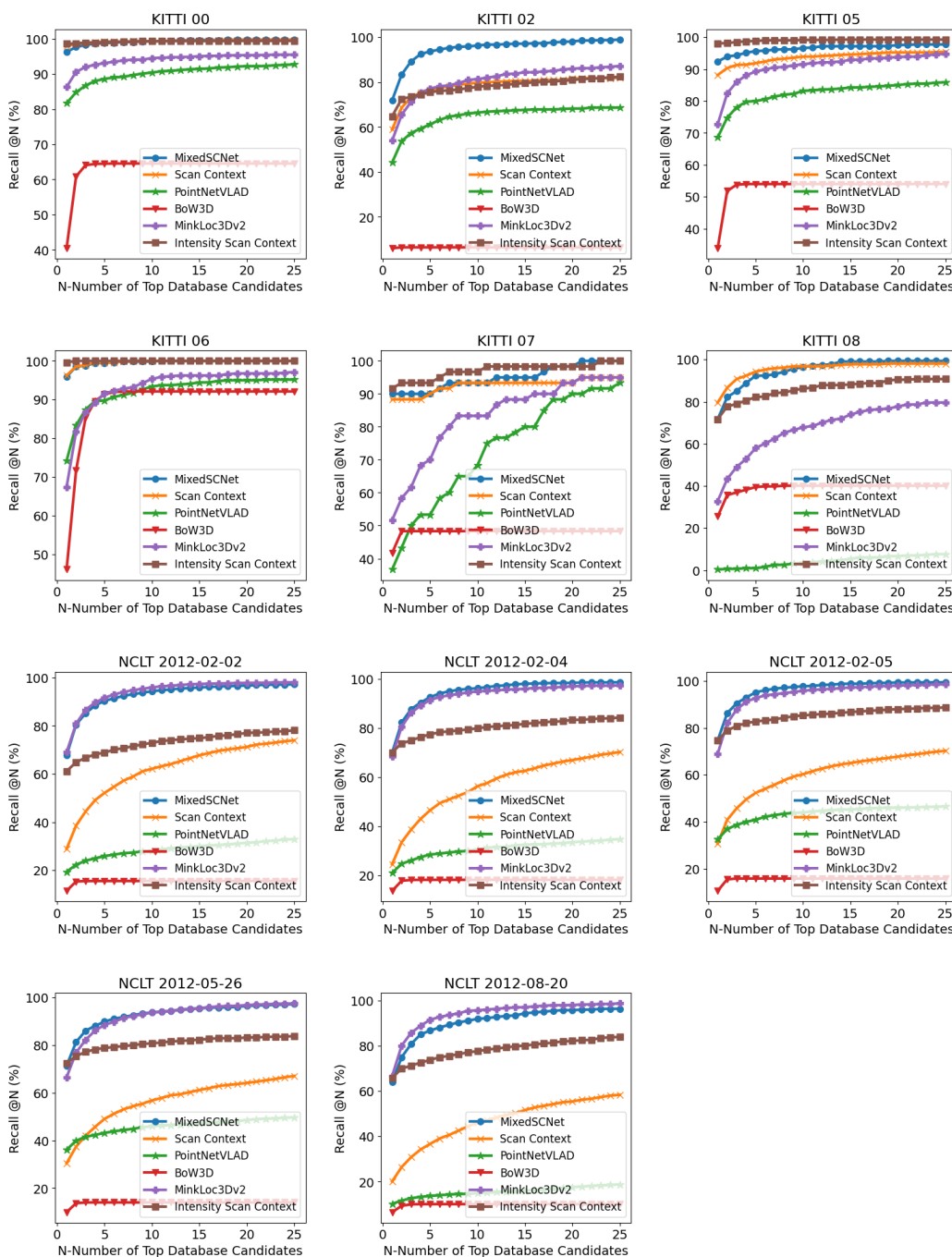

**Figure 6.** Recall–candidate number curve for all sequences.

## 4.4. System Runtime

Table 3 presents the average runtime of the six methods on the KITTI 07 sequence, with all tests having been conducted on a computer equipped with an AMD Ryzen 7 5825U CPU ( AMD, Wuhan, China). It is evident that excluding Intensity Scan Context, the majority of the processing time for other methods is allocated to the generation of global descriptors, while the database retrieval time remains relatively short. Intensity Scan Context, due to the necessity of computing both geometric and intensity scores for each candidate frame, shows the longest database retrieval time, which grows linearly with database size. To ensure fairness, all methods are tested on a CPU, and due to the utilization of neural networks in our method, PointNetVLAD, and MinkLoc3Dv2, the descriptor generation time is relatively extended for these three methods. If GPU testing were employed, the runtime would significantly decrease.

**Table 3.** Average runtime of all methods on KITTI 07.

| Method | Descriptor Generation (ms) | Database Retrieval (ms) |
|---|---|---|
| PointNetVLAD | 203.73 | 0.40 |
| Scan Context | 2.45 | 0.10 |
| Intensity Scan Context | 12.57 | 68.12 |
| MinkLoc3Dv2 | 219.01 | 4.42 |
| BoW3D | 15.08 | 2.53 |
| MixedSCNet (ours) | 50.99 | 1.56 |

Furthermore, the descriptor generation time for MixedSCNet is approximately $\frac{1}{4}$ of that for MinkLoc3Dv2. This discrepancy is attributed, firstly, to the more streamlined architecture of MixedSCNet, resulting in reduced computational complexity. Secondly, MinkLoc3Dv2 incorporates ground fitting and downsampling operations during point cloud preprocessing, whereas MixedSCNet directly processes the point cloud, solely computing the maximum height, intensity, and smoothness information within each sub-region. Consequently, MixedSCNet exhibits an overall lower computational load. It is shown that MixedSCNet exhibits the shortest overall runtime among the three top-performing methods.

**5. Conclusions**

This paper presents a place recognition method based on Bird's Eye View (BEV) descriptors and deep learning. By leveraging computationally efficient handcrafted descriptors and a lightweight neural network, the proposed approach achieves rapid place recognition. Additionally, it demonstrates robust adaptability to reverse visit situations and superior generalization performance in scenarios involving cross-domains. Initially, the method transforms point clouds into a three-channel BEV descriptor, termed MixedSC, which encodes height, intensity, and smoothness information. Subsequently, a compact deep neural network, MixedSCNet, is trained to transform MixedSC into a global descriptor enriched with high-level features. At last, the KD tree facilitates the retrieval of multiple candidate frames from the database based on their similarity to the query frame.

Experimental results indicate that MixedSCNet achieves commendable performance on both the KITTI and NCLT datasets. Compared with suboptimal methods, such as Intensity Scan Context and MinkLoc3Dv2, MixedSCNet exhibits consistent performance across two different datasets with the lowest computational cost. It is noteworthy that the outstanding performance of MixedSCNet persists even on the NCLT dataset, which exhibits a huge difference from the training domain, underscoring the robustness and generalization capabilities of the proposed method.

There are a few limitations to our methods, and future work will try to fix them. Firstly, the way we combine MixedSC and the neural network is not end-to-end and takes two steps to solve the place recognition task. Secondly, the neural network architecture employed in this study could benefit from further refinement, e.g., adding attention mechanisms. Lastly, in comparison to methods based on manually crafted descriptors, the speed of the proposed approach in this paper still requires improvement. It is worthwhile to explore the utilization of lighter network architectures to enhance the recognition speed.

**Author Contributions:** Conceptualization, T.S. and Y.S.; methodology, Y.S.; software, Y.S.; validation, Y.S. and W.H.; formal analysis, Y.S. and W.H.; investigation, Y.S., D.Y. and B.B.; resources, Y.S., J.D., X.Z. and T.S.; data curation, Y.S., D.Y. and B.B.; writing—original draft preparation, Y.S. and W.H.; writing—review and editing, Y.S., W.H. and T.S.; visualization, Y.S. and W.H.; supervision, T.S.; project administration, J.D., X.Z. and T.S. All authors have read and agreed to the published version of the manuscript.

**Funding:** This research was funded by the Key Research and Development Plan Project of Hubei Province under grant 2021BAA196.

**Data Availability Statement:** Data is contained within the article.

**Conflicts of Interest:** The authors declare no conflicts of interest.

## Abbreviations

The following abbreviations are used in this manuscript:

| | |
|---|---|
| LiDAR | Light Detection and Ranging |
| BEV | Bird's Eye View |
| MixedSC | Mixed Scan Context |
| KITTI | Karlsruhe Institute of Technologyand Toyota Technological Institute |
| NCLT | North Campus Long-Term |

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
