# Peer review of "MixedSCNet: LiDAR-Based Place Recognition Using Multi-Channel Scan Context Neural Network"

_electronics, doi:10.3390/electronics13020406_

Round 1
Reviewer 1 Report
Comments and Suggestions for Authors
The paper proposes a place recognition method based on a LiDAR point cloud. The proposed method extends the Scan Context and Intensity Scan Context methods by providing additional information about the local point cloud structure (smoothness) and using 2D CNN to generate a discriminative global descriptor from 2D scan context map. There's limited novelty in the proposed approach - some other methods, such as DiSCO [A], also use neural networks to generate a global descriptor from Scan Context-based maps.
The paper is relatively well-written and logically organized. The Related Work section discusses the relevant methods.
The proposed method is described with sufficient details.
The biggest problem is that the performance of the proposed method is compared with very weak and relatively old methods such as PointNetVLAD or ScanContext.
PointNetVLAD, proposed in 2018, was the first neural network-based method to generate a global point cloud descriptor. Since then, many better methods have been proposed, outperforming PointNetVLAD by a large margin. For example, PointNetVLAD has 63.3% Recall@1 on Oxford RobotCar dataset (in so-called "refined" evaluation scenario), whereas MinkLoc3Dv2 [B] has 96.9. Several other methods (e.g. TransLoc3D [C], SVT-Net[D]) also scored above 90% in the same benchmark. That's almost 30 p.p. better than PointNetVLAD.
Also, several improvements to ScanContect were proposed: Intensity Scan Context, Disco, and ScanContext++ with significantly improved performance.
I recommend authors to improve the experimental evaluation section, by experimentally comparing their method with up-to-date baselines, such as TransLoc3D, MinkLoc3Dv2, Intensity ScanContext or ScanContext++. Source code and trained models of many of these methods are publicly available.
=====
References:
[A] X. Hu et al. DiSCO:Differentiable Scan Context with Orientation, 2021
[B] MinkLoc++: Lidar and Monocular Image Fusion for Place Recognition, 2021
[C] TransLoc3D: Point cloud based large-scale place recognition using adaptive receptive fields, 2021
[D} Svt-net: Super light-weight sparse voxel transformer for large scale place recognition, 2022
Comments on the Quality of English LanguageThe paper is relatively well-written and logically organized. Minor language issues - some sentences or words sound unnatural, e.g.
"A training sample can be externalized as a tuple..." 'externalized' doesn't sound like a correct word here.
Rather 'A training sample can be represented as a tuple...'
Author Response
Dear Reviewer,First of all, thank you for your detailed comments and suggestions! I'm very grateful for your time and effort in reviewing our paper.
Firstly, addressing your concern regarding the "limited novelty" of our approach, it is acknowledged that the proposed methodology shares similarities with DiSCO. However, it is important to note that DiSCO's process for extracting global descriptors involves computationally intensive steps. DiSCO initially transforms the point cloud into a Bird's Eye View (BEV) image, followed by a conversion to polar coordinates. Subsequently, a U-Net is employed to extract feature maps, and a Fourier transform is applied to transform these maps into the frequency domain, ultimately yielding the spectral representation as the global descriptor. Similarly, MinkLoc3Dv2 involves point cloud preprocessing, including downsampling and ground removal, followed by neural network inference, making the overall computation time considerable. Notably, both DiSCO and MinkLoc3Dv2 exhibit a significant drop in recall at the top 1% when applied to different datasets, indicating the need for improved generalization. The primary innovation of our proposed MixedSCNet lies in its simplicity and effectiveness: incorporating smoothness as a dimension in the BEV descriptor and utilizing a straightforward network architecture to achieve state-of-the-art performance within a shorter computation time while maintaining robust generalization across diverse test sets.
Regarding your second question on comparing with up-to-date baselines, I attempted to reproduce the results of the four methods you recommended: MinkLoc3Dv2, Intensity Scan Context, TransLoc3D, and SVT-Net. Successfully, I managed to reproduce the results for the first two methods, and the updated paper now reflects these outcomes. However, I encountered difficulties in reproducing the results for TransLoc3D, as it only provided source code without accompanying model weights. Additionally, the provided SVT-Net model weights did not align with the model structure in its source code, making it impossible to load the weights.
Concerning your third question about minor language issues, I have revised the statements in the paper and conducted a thorough proofreading.
Again, thanks for your generous corrections!
Best regards, Mr. Si
Reviewer 2 Report
Comments and Suggestions for Authors
This paper proposes a lidar-based place recognition method utilizing Bird’s Eye View (BEV) descriptors and lightweight neural networks. The reviewer hopes to hear responses from the authors regarding the following comments:
1. The idea of combining BEV feature representation with deep neural networks is not new. What is the unique invention of the proposed method compared to existing methods?
2. Further explanatory words are needed to justify the novelty of the proposed MixedSCNet. It seems that MixedSCNet is quite similar to the widely used ResNet series. What is the unique advantage of the proposed MixedSCNet?
3. Hyperparameter settings are not explained in detail.
4. In the experimental study section, in addition to recall plots, the authors should also provide qualitative graphical results to better illustrate the recognition performance of different methods for cross comparison purposes.
5. What is the influence of the number of laser channels (i.e., 32-line vs 64-line) of lidar sensors on the recognition results by the proposed method?
6. What, if any, are the limitations of the proposed method?
Comments on the Quality of English LanguageOK.
Author Response
Dear Reviewer,First of all, thank you for your detailed comments and suggestions! I'm very grateful for your time and effort in reviewing our paper.
1. Firstly let me answer your question about the unique invention of our proposed method. It is acknowledged that the proposed methodology shares similarities with DiSCO. However, it is important to note that DiSCO's process for extracting global descriptors involves computationally intensive steps. DiSCO initially transforms the point cloud into a Bird's Eye View (BEV) image, followed by a conversion to polar coordinates. Subsequently, a U-Net is employed to extract feature maps, and a Fourier transform is applied to transform these maps into the frequency domain, ultimately yielding the spectral representation as the global descriptor. Similarly, MinkLoc3Dv2 involves point cloud preprocessing, including downsampling and ground removal, followed by neural network inference, making the overall computation time considerable. Notably, both DiSCO and MinkLoc3Dv2 exhibit a significant drop in recall at the top 1% when applied to different datasets, indicating the need for improved generalization. The primary innovation of our proposed MixedSCNet lies in its simplicity and effectiveness: incorporating smoothness as a dimension in the BEV descriptor and utilizing a straightforward network architecture achieve state-of-the-art performance within a shorter computation time, while maintaining robust generalization across diverse test sets.
2. Regarding your second question, MixedSCNet shares a similar structure with ResNet18, with variations only in the convolutional kernel size and feature dimensions. The network structure itself is not the core focus of this paper; ResNet18 was chosen as the backbone due to its simplicity, ease of convergence, and lower computational cost during inference.
3. Addressing your third question, the hyperparameters have been detailed and highlighted in the revised paper in line 369-380.
4. Concerning your fourth question, qualitative experimental results have been added to the revised paper in line 466-475 and figure 4, 5.
5. Regarding your fifth question, this study utilized two datasets: the KITTI dataset with a 64-line LiDAR and the NCLT dataset with a 32-line LiDAR. The method was trained on partial sequences from KITTI and tested on the remaining KITTI sequences and the NCLT dataset. Experimental results demonstrate a top 1% recall rate of 99.30% on KITTI and 98.98% on NCLT, indicating that the proposed method is basically not that sensitive to the number of LiDAR lines.
6. Addressing your sixth question, the limitations of our method primarily include: 1. Non-end-to-end nature - the neural network takes BEV descriptors as input instead of raw point clouds. 2. The neural network structure requires further improvement, and the introduction of attention mechanisms could enhance performance. 3. In comparison to methods based on manually designed descriptors, the speed of our method still needs improvement, and consideration could be given to employing more lightweight network architectures. Above three limitations are mentioned in the conclusion section of our revised paper.
Again, thanks for your generous corrections!
Best regards, Mr. Si
Round 2
Reviewer 1 Report
Comments and Suggestions for Authors
The revised version addresses my concerns after the review of the first version of the manuscript. I recommend publishing the paper.